# Inhibition of the Cell Uptake of Delta and Omicron SARS-CoV-2 Pseudoviruses by *N*-Acetylcysteine Irrespective of the Oxidoreductive Environment

**DOI:** 10.3390/cells11203313

**Published:** 2022-10-21

**Authors:** Sebastiano La Maestra, Silvano Garibaldi, Roumen Balansky, Francesco D’Agostini, Rosanna T. Micale, Silvio De Flora

**Affiliations:** 1Department of Health Sciences (DISSAL), University of Genoa, 16132 Genoa, Italy; 2Department of Internal Medicine and Medical Specialties (DIMI), University of Genoa, 16132 Genoa, Italy; 3National Centre of Oncology, 1756 Sofia, Bulgaria

**Keywords:** SARS-CoV-2, COVID-19, cell internalization, *N*-acetyl-L-cysteine, ascorbic acid, hydrogen peroxide, reactive oxygen species

## Abstract

The binding of SARS-CoV-2 spikes to the cell receptor angiotensin-converting enzyme 2 (ACE2) is a crucial target both in the prevention and in the therapy of COVID-19. We explored the involvement of oxidoreductive mechanisms by investigating the effects of oxidants and antioxidants on virus uptake by ACE2-expressing cells of human origin (ACE2-HEK293). The cell uptake of pseudoviruses carrying the envelope of either Delta or Omicron variants of SARS-CoV-2 was evaluated by means of a cytofluorimetric approach. The thiol *N*-acetyl-L-cysteine (NAC) inhibited the uptake of both variants in a reproducible and dose-dependent fashion. Ascorbic acid showed modest effects. In contrast, neither hydrogen peroxide (H_2_O_2_) nor a system-generating reactive oxygen species (ROS), which play an important role in the intracellular alterations produced by SARS-CoV-2, were able to affect the ability of either Delta or Omicron SARS-CoV-2 pseudoviruses to be internalized into ACE2-expressing cells. In addition, neither H_2_O_2_ nor the ROS generating system interfered with the ability of NAC to inhibit that mechanism. Moreover, based on previous studies, a preventive pharmacological approach with NAC would have the advantage of decreasing the risk of developing COVID-19, irrespective of its variants, and at the same time other respiratory viral infections and associated comorbidities.

## 1. Introduction

The initiating step of the infection caused by severe acute respiratory syndrome coronavirus 2 (SARS-CoV-2), an enveloped single-stranded RNA virus of the genus Betacoronavirus1, is the binding of the virus to susceptible cells and its subsequent cell uptake to trigger the intracellular replication process. Hence, this is a major target both in the prevention and in the etiologic therapy of coronavirus disease 2019 (COVID-19). Several mechanisms can be involved in this process, the most important being the binding of virus spikes to the cell receptor angiotensin-converting enzyme 2 (ACE2 or peptidyldipeptidase A) and subsequent membrane fusion [1]. The SARS-CoV-2 spike contains 14 disulfide (S-S) bonds [2]. It is made of two subunits, of which S1 contains the receptor-binding domain (RBD) and S2 is responsible for membrane fusion [3]. This protease is an integral membrane protein detected in epithelial, endothelial, and myocardial cells, as well as in T lymphocytes, macrophages, and hepatocytes [4,5]. ACE2 is involved in the renin/angiotensin system together with the angiotensin-converting enzyme (ACE), which has opposite effects. In fact, while ACE causes vasoconstriction, inflammation, apoptosis, and oxidative stress due to the production of reactive oxygen species (ROS), ACE2 causes vasodilatation, angiogenesis, antioxidative, and antiapoptotic effects [6]. Therefore, the binding of SARS-CoV-2 to ACE2 not only allows virus penetration into cells but also causes a deprivation of this protective mechanism. The reduced catalytic efficiency of ACE2 resulting from viral binding and the downregulation of this receptor appears to be detrimental in patients with a baseline deficiency of the ACE2 receptor activity, including patients with advanced age, cardiovascular risk factors, and previous cardiovascular events [7].

Both the RBD on the spike domain and ACE2 have several cysteine residues, and the binding affinity is decreased when the disulfide bonds (S-S) of ACE2 and SARS-CoV-2 spike proteins are reduced to sulfhydryl groups (SH). Therefore, the redox environment of cell surface receptors is regulated by the thiol–disulfide equilibrium in the extracellular region [8]. This is not peculiar for SARS-CoV-2, since many other viruses use the thiol–disulfide exchange to initiate cell entry [9]. Furthermore, several experimental studies, using a variety of methodologies, have suggested that thiols may exert inhibitory effects on the binding of SARS-CoV-2 to ACE2 [10,11,12,13,14,15,16].

Oxidative stress represents a major mechanism in the pathogenesis, not only of a variety of chronic degenerative diseases, among which is cancer [17], but also of infectious diseases, also including viral respiratory diseases. Similar to the influenza virus infection and other viruses, coronavirus infection affects the oxidative stress machinery, enhancing ROS production and the weakening of defense mechanisms [18]. Overproduction of both ROS, such as the superoxide anion (O_2_^−^) and the hydroxyl radical (^•^OH), and nonradical reactive molecules, such as hydrogen peroxide (H_2_O_2_), are involved in the pathogenesis, progression and severity of SARS-CoV and SARS-CoV-2 [19]. In addition, these species cause redox-modulated signaling cascades mediated by transcription factors, such as activator protein-1 (AP-1), nuclear factor kappa-light-chain-enhancer of activated B cells (NF-κB), and nuclear factor erythroid 2-related factor 2 (Nrf2) [20]. Due to the negative role played by the deprivation of antioxidant systems, antioxidants are expected to exert protective effects towards this infection as well as towards associated comorbidities [21].

While the role of the thiol–disulfide balance as a key player in SARS-CoV-2 internalization into cells is well established, less is known about the involvement of oxidoreductive mechanisms in the extracellular environment. These premises prompted us to explore the effects of oxidants and antioxidants on virus uptake by susceptible cells. To this purpose, we used an ACE2 expressing cell line of human origin, which was challenged with SARS-CoV-2 pseudoviruses derived from either the Delta or Omicron variants, whose internalization was evaluated by means of a cytofluorimetric approach. Pseudoviruses are lentiviruses belonging to the family of retroviridae that integrate the envelope glycoprotein of SARS-CoV-2 to form a virus with an exogenous viral envelope, whereas the genome retains the characteristics of the retrovirus itself. The pseudovirus is not able to duplicate because it lacks the coronavirus genetic patrimony.

The investigated antioxidants included *N*-acetyl-L-cysteine (NAC) and ascorbic acid (AsA, vitamin C). NAC is an analogue and precursor of the tripeptide reduced glutathione (GSH, gamma-glutamylcysteinylglycine), a formidable barrier against toxic and infectious agents. NAC easily penetrates cells where it is deacetylated to yield L-cysteine (L-Cys), the only naturally occurring amino acid that carries a thiol-containing side chain. L-Cys is the rate-limiting substrate for GSH biosynthesis, which is mainly achieved through the activation and upregulated production of glutamate-cysteine ligase (GCL) [22]. Replenishment of depleted GSH stores occurs both by GSH recycling and by de novo synthesis of this tripeptide. Accordingly, the rescue of GSH through NAC is a treatment strategy for a broad array of different diseases sharing a pathogenetically relevant loss of GSH [23]. It was found that the depletion of thiols in the blood serum is the best indicator capable of discriminating patients with severe COVID-19 forms requiring hospitalization in intensive care units [24].

It is noteworthy that SARS-CoV-2, alike with other viral infections, such as HIV, influenza, and HSV [25], promotes an oxidized environment in the host cell and markedly decreases the levels of total cellular thiols and of GSH in infected cells, and it stimulates the oxidation of GSH to oxidized glutathione (GSSG), an effect that is counteracted by NAC [26]. The extracellular efflux of cellular thiols is also reduced, which may support the hypothesis that SARS-CoV-2 infection promotes a pro-oxidant environment, thereby interfering with the cystine–cysteine cycle of the cell, and thus with the redox homeostasis mechanisms of extracellular thiols [25]. NAC mainly works as a scavenger of ROS, and AsA is a water-soluble reducing and antiradical agent acting by electron transfer reactions. The agents used to create an oxidizing environment included either H_2_O_2_ or a mixture of hypoxanthine (HX) with xanthine oxidase (XO). H_2_O_2_ is widely regarded as a cytotoxic agent whose levels must be minimized by the action of antioxidant defense enzymes [27]. XO catalyzes the oxidation of HX to xanthine (X) and can further catalyze the oxidation of X to uric acid, which reduces O_2_ to O_2_^−^. This eventually progresses to H_2_O_2_, forming ^•^OH in the presence of a transition metal.

After having documented the ability of NAC to inhibit the penetration of SARS-CoV-2 pseudoviruses into ACE2-expressing cells, a further goal of the present study was to investigate whether the simultaneous exposure of cells to NAC and either AsA or H_2_O_2_ or the ROS-generating system may alter this protective effect. The rationale for these experiments relies on the known interactions between the agents tested. In fact, a close relationship between AsA and GSH was predicted soon after the characterization of the chemical formulae of the two molecules almost one century ago and, according to the “Foyer-Halliwell-Asada” pathway, these two agents are defined as the “heart of the redux hub”. In particular, oxidation of GSH to GSSG is linked to the reduction of ascorbate to dehydroascorbate and the generation of H_2_O_2_ and O_2_^−^ [28]. The results of these experiments led to the conclusion that inhibition of the cell uptake of SARS-CoV-2 pseudoviruses by NAC is not affected by alterations of the extracellular microenvironment by oxidants and antioxidants.

## 2. Materials and Methods

### 2.1. Cells

The ACE2-HEK293 cell line was purchased from Tebu-bio (Magenta, Milano, Italy). These cells are recombinant clonal stable Human Embryo Kidney cell lines (HEK293) that constitutively express full-length human ACE2. The HEK293 cell line was originally derived from human embryonic kidney cells grown in tissue culture taken from an aborted female fetus and was immortalized by means of the Large-T antigen (L-T) of SV40 polyomavirus. L-T has a transforming capacity because it can bind the p53 and pRB oncosuppressor genes and inactivate them. Therefore, HEK293 cells are transfected with the gene encoding ACE2 in such a way to express the protein stably. The cells were maintained and cultured by using Dulbecco’s Modified Eagle Medium (DMEM) containing 10% fetal bovine serum (FSB), 2 mM L-glutamine, 1% nonessential amino acids, 1 mM Na-pyruvate, 1% penicillin/streptomycin, and 0.5 μg/mL puromycin at a temperature of 37 °C in a humidified atmosphere of 5% CO_2_. Moreover, transfected cells were selected and maintained by adding hygromycin (100 μg/mL) and puromycin (1 μg/mL) in a complete medium. Cell cultures were trypsinized when they reached approximately 70–80% density in the monolayer and used for subcultures or seeded in 96-well polystyrene tissue culture microplates (Nest Scientific, Beltsville, MD, USA) at a density of 1 × 10^4^ cells per well for subsequent testing. Since these cells do not adhere easily to the polystyrene surface and tend to detach when manipulating the cultures, α-polylysine (0.1 mg/mL) (SERVA, Heidelberg, Germany) was added as an attachment factor that improves cell adherence based on the interaction between the positively charged polymer and negatively charged cells.

### 2.2. Viruses

The pseudoviruses derived from the Delta and Omicron variants of concern (VOC), according to the WHO classification, were from BPS Bioscience (San Diego, CA, USA). They contain the enhanced Green Fluorescence Protein (eGFP) reporter driven by a CMV promoter. The SARS-CoV-2 variant B1.617.2, also known as the Delta variant, was first identified and documented in India in October 2020. This variant has the deletion E156/F157 and several mutations in the spike protein (T19R, G142D, R158G, L452R, T478K, D614G, P681R, D950N) that may lead to higher transmissibility and infectivity. The virus stock used (Lot 220228) had a declared titer of 5 × 10^6^ TU (Transducing Units)/mL. The SARS-CoV-2 variant B1.1.529, also known as the Omicron variant, was identified in South Africa in November 2021, and thereafter its circulation was documented in several countries, having been detected in 110 countries as of 23 December 2021 [29]. This variant has a large number of spike mutations (A67V, Δ69–70, T951, G142D, Δ143–145, Δ211, L2121, ins214EPE, G339D, S3711, S373P, S375F, K417N, N440K, G446S, S477N, T478K, E484A, Q498R, N501Y, Y505H, T547K, T614G, H665Y, N679K, P681H, N764K, D769Y, N856K, Q954H, N969K, L981F) that allow the virus to spread more readily compared with other variants. The virus stock used (Lot 220114) had a declared titer of 3 × 10^6^ TU/mL.

### 2.3. Test Compounds and ROS Generating System

NAC was a gift by Zambon Group (Vincenza, Italy) as a pure compound in the form of a mesh powder. AsA, H2O2, HX, and XO were purchased from Sigma Aldrich (Merck Life Science, Milan, Italy). To measure the efficacy of the reaction between XO and HX to generate ROS, we have developed a simple method using a free radical probe, dichlorofluorescindiacetate (DCFH-DA). DCFH-DA is able to recognize radical species in a cellular system after it enters the intracellular environment. In the cytosol, cellular esterases hydrolyze DCFH-DA to DCFH, which is then oxidized by free radicals to the fluorescent DCF. We mixed XO and HX at different concentrations and added 1% of human serum to ensure the presence of esterases. Subsequently, fluorescence emission was measured by fluorometric analysis at 535 nm emission and 485 nm excitation wavelengths. The results were expressed as the increase in fluorescence compared to control samples missing HX and XO.

### 2.4. Cytotoxicity of Oxidants and Antioxidants

In order to assess the toxicity in ACE2-HEK293 cells, solutions of test compounds were tested using the methyltetrazolium (MTT assay). However, given the reducing action exerted by NAC on the yellow tetrazolium salt, the subtoxic doses for NAC were evaluated using propidium iodide (PI). This nonpermeable fluorescent agent binds to DNA by intercalating between the bases with an excitation maximum of 535 and an emission maximum of 617 nm. To this purpose, cell monolayers at 80% confluence were detached by trypsin/EDTA treatment, and the cell suspensions were seeded in 96-well poly-l-lysine coated culture plates (1 × 10^4^ cells/well). After 24 h, the cells were treated with varying concentrations of freshly prepared oxidants or antioxidants for 24 h. Thereafter the medium was removed and the cell monolayers were rinsed three times with PBS. Finally, the cells were supplemented with PI at 10 µg/mL (Invitrogen, Waltham, MA, USA) and incubated for 15 min at 37 °C in a 5% CO_2_ atmosphere. After incubation, the cell fluorescence was measured in a microplate reader (Spark Multimode Reader, Tecan, Switzerland) at 485 nm emission and 595 nm excitation wavelengths.

### 2.5. Internalization of Pseudoviruses into ACE2 Expressing Cells

In order to evaluate the internalization of pseudoviruses into ACE2-HEK293 cells and its possible modulation by oxidants and antioxidants, we used cytofluorimetric analyses. Briefly, 1 × 10^4^ cells were seeded in each well into 96-well plates. After 24 h, 5 µg/mL of polybrene (1,5-dimethyl-1,5-diazaundecamethylene polymethobromide, Hexadimethrine bromide) (Tebu-bio, Magenta, Milano, Italy) were added to each well in order to enhance the efficiency of the lentiviral infection. Subsequently, each well received a specific treatment, such as inoculation of varying amounts of oxidants or antioxidants and/or varying doses, expressed as TU (transduction units), of SARS-CoV-2 spike pseudotyped lentiviruses. Fifty-six hours later, the cells were trypsinized and resuspended in 300 µL of medium to detect the transduction of the expression of eGFP in the target cells by cytofluorimetric analysis. Flow cytometry was performed using a Navios flow cytometer and Kaluza analysis software 2.1 (Beckman Coulter, Brea, CA, USA).

### 2.6. Statistical Analysis

The results were expressed as means ± SD of triplicates either of the virus internalization or of the percent internalization compared with the virus control in the absence of test compounds. Comparison between two groups was made by Student’s *t* test. The *p* values < 0.05 were regarded as statistically significant.

## 3. Results

### 3.1. Toxicity of Oxidants and Antioxidants in ACE2 Expressing Cells

A series of preliminary experiments were carried out in order to optimize the solubilization conditions of test compounds and their toxicities to ACE2-HEK293 cells, over a broad range of concentrations, as well as the methodology used for evaluating cell viability. Since NAC is acidic due to its carboxyl group (pKa = 3.14), and due to the absence of a free amino group, it is necessary to buffer its solutions. To this purpose, NAC was solubilized in MEM and adjusted to pH 7.2–7.4 with NaOH. AsA, H_2_O_2_, and the mixture of HX with XO did not need to be buffered when diluted in MEM.

The results of final cytotoxicity assays using a narrow range of compound concentrations are shown in Figure 1. The assays were conducted to identify the subtoxic concentrations in order to avoid toxic effects in the cellular model used. Cell viability was determined after 24 h of treatment with a range of antioxidant/oxidant concentrations. The concentrations of test compounds that decreased viability by at least 30% were considered to be toxic. By following this criterion, AsA was toxic at concentrations above 200 µM, H_2_O_2_ above 50 µM, and HX/XO above 0.5 mM/0.03 mU. NAC was toxic only at the highest concentration tested (300 mM). However, we decided to use NAC at 25 mM because, at this concentration, the cells did neither show any cytopathic effect nor loss of adhesiveness.

### 3.2. Inhibition of Cell Internalization of Delta SARS-CoV-2 Pseudovirus by NAC

Figure 2 shows a cluster of ACE2-HEK293 cells. The green cells in the right panel were those in which the delta SARS-CoV-2 pseudovirus had been internalized, as demonstrated by an enhanced expression of green fluorescent protein (eGFP).

In a series of experiments the dose-related efficiency of the delta pseudovirus used to infect the ACE2-bearing cells was assessed either in the absence or in the presence of a fixed dose of NAC (25 mM). The panels on the left in Figure 3A provide evidence for the high efficiency of the test system used, with the internalization of the virus affecting the vast majority of exposed cells at the highest virus dose tested. The panels on the right show the remarkable protective capacity of the drug at a glance, with a sharp decrease of green dots that approaches 0% at the lowest virus dose.

The comparison between NAC-free and NAC-treated cells is summarized in Figure 3B, showing the means of three replicates with SD < 10% of the means. The following conclusions can be drawn: (*a*) The proportion of internalized pseudovirus is related to the dose of virus challenged with cells, with 25.1% of cells infected by the virus at the dose of 0.5 × 10^4^ TU, 31.0% at the dose of 1 × 10^4^ TU, and 78.1% at the dose of 2.5 × 10^4^ TU; (*b*) At all virus doses, a fixed concentration of NAC (25 mM) significantly and considerably inhibits the proportion of internalized virus; (*c*) Inhibition by NAC is inversely related to the dose of virus. In particular, the inhibition coefficients at the doses of 0.5, 1, and 2.5 × 10^4^ TU of virus, calculated as [(internalized virus in the absence of NAC -internalized virus in the presence of NAC)/internalized virus in the absence of NAC × 100] are 81.4%, 62.7%, and 42.2%, respectively.

Thereafter, we evaluated the inhibition of a fixed dose of Delta virus (10,000 TU) by concentrations of NAC varying between 1.56 and 25 mM. Examples of the results obtained at cytofluorimetric analyses are shown in Figure 4A, whereas Figure 4B shows graphically, on an arithmetic scale, the dose-dependence of the protective effect of NAC. There was a high and statistically significant (*p* < 0.001) inverse correlation between the concentration of NAC (x) and the proportion of SARS-CoV-2 pseudovirus (y) internalized into ACE2-HEK293 cells (R^2^ = 0.9808). The regression line equation is y = 1.5177x + 60.085, from which it can be inferred that a 50% inhibition of virus internalization would be expected to occur, at that virus dose, at a NAC concentration of 6.6 mM and a total inhibition would be achieved at a NAC concentration of 39.6 mM.

### 3.3. Effect of Oxidants and Antioxidants on Cell Internalization of Delta and Omicron SARS-CoV-2 Pseudoviruses

A fixed dose of either Delta or Omicron SARS-CoV-2 pseudoviruses (10,000 TU/culture) was challenged with ACE2-HEK293 cells, either untreated (control) or exposed to antioxidant agents, including NAC (25 mM) and ascorbic acid (AsA, 200 µM), or oxidants, including hydrogen peroxide (H_2_O_2_, 50 µM) and a mixture of hypoxanthine (HX 31 µM) with xanthine oxidase (XO, 0.5 mU/mL). As assessed by means of the DFC method, the HX/XO system generated ROS, with a dose-dependent increase of the fluorescent signal (Figure 5).

The results of the experiments evaluating internalization of delta and omicron SARS-CoV-2 pseudoviruses (10,000 TU/culture) into ACE2-HEK293 cells are summarized in Figure 6, where the data recorded in the presence of either oxidants or antioxidants are expressed as the percent of the values recorded in virus controls. In the absence of any test compounds, a high proportion of the virus was internalized into the cells following a load of 10,000 TU/culture. Neither H_2_O_2_ nor HX + XO significantly affected the efficiency of virus internalization per se. AsA caused a small but statistically significant (*p* < 0.05) inhibition of cell uptake of both viruses. NAC significantly (*p* < 0.001) and substantially decreased the cell uptake of both viruses when tested alone. In addition, this thiol exhibited a protective effect of the same order of magnitude when challenged in combination either with oxidants (H_2_O_2_ or HX + XO) or other antioxidants working with a different mechanism (AsA).

## 4. Discussion

Neither H_2_O_2_ nor the ROS-generating system used were able to affect the ability of either the Delta or Omicron SARS-CoV-2 pseudoviruses to be internalized into ACE2-expressing cells. In contrast, the reducing agents NAC and, to a milder extent, AsA exerted protective effects on this crucial step of the virus infection. Inhibition of the cell uptake of the viruses by NAC occurred at nontoxic doses, it was reproducible in several experiments, and it was dose-dependent. Importantly, it was of the same order of magnitude in the two virus variants under study. In addition to defending susceptible cells towards virus infectivity, hampering the virus–receptor interaction is expected to preserve the integrity of ACE2, which plays a protective role [6]. In particular, a reduced expression of the enzyme causes angiotensin II accumulation generated by ACE and the depleted ACE2 is unable to convert angiotensin I into the vasodilator heptapeptide angiotensin, thus generating pulmonary injury [30].

Our experimental results support the computational findings of Hati and Bhattacharyya [8], who postulated that the redox environment of cell surface receptors is regulated by the thiol–disulfide equilibrium in the extracellular region. Other laboratories, using different methodological approaches and with variable results, evaluated the ability of thiols to hamper the uptake either of SARS-CoV-2 or of SARS-CoV-2 pseudoviruses by ACE2-expressing cells. Grishin et al. [11] found that dithiothreitol (DTT) and tris(2-carboxyethyl)phosphine (TCEP) and, to a lower extent, NAC and GSH produced a significant decrease in the affinity of the RBD for ACE2. Moreover, DTT and TCEP decreased the infectivity of SARS-CoV-2 in Vero’76 cells, whereas no antiviral effect of either GSH or NAC was detectable due to decreases in viability in those cells. The in vitro toxicity of NAC observed in these experiments contrasts with the low toxicity exhibited by buffered NAC in the present study as well as with the recognized safety of NAC-containing pharmaceutical preparations in clinical use. For instance, intravenous NAC is extensively used at doses as high as 150 mg/kg b.w (more than 10 g in a 70 kg man), which are the standard regimen as a clinically approved antidote against paracetamol (acetaminophen) intoxication [31]. On the other hand, Murae et al. [15] found that both NAC and GSH reduced the spike-mediated cell–cell fusion and syncytium formation in VeroE6/TMPRSS2 cells. Moreover, as assessed by luciferase assay, both NAC and GSH dose-dependently inhibited the entrance of pseudoviruses carrying the spike proteins of SARS-CoV-2 into the same cells. Debnath et al. [14] showed that NAC forms covalent conjugates with cysteine residues of spike protein that were disulfide bonded in the native state. Moreover, NAC inhibited the SARS-CoV-2 replication in VeroE6 cells, as assessed by real-time PCR. Conversely, Manček-Keber et al. [12] found that NAC amide, but not NAC or GSH, was able to prevent the fusion of HEK293T spike-expressing cells, as evaluated by measuring luciferase activity. Khanna et al. [13] found that multiple thiol drugs inhibit SARS-CoV-2 binding to ACE2 and virus entry into cells, whereas the intraperitoneal injection of cysteamine to hamsters infected with SARS-CoV-2 decreased inflammation in the lungs but failed to decrease viral infection. In a further study, the thiol-based chemical probes P2119 and P2165 inhibited infection by human coronaviruses, including SARS-CoV-2, and decreased the binding of spike glycoprotein to ACE2 [16]. Cotreatment with NAC and other agents was also investigated. Akhter et al. [10] tested a combination of NAC with Bromelain (Brom). As assessed by UV spectral detection, NAC reduced the disulfide bonds in the recombinant SARS-CoV-2 spike protein and envelope proteins. Individually, neither NAC nor Brom inhibited the cytopathic effect induced both by wild-type and mutant strains of SARS-CoV-2 in Vero cells, whereas their combination (BromAC) exerted protective effects.

In our study, AsA showed a small but statistically significant ability to inhibit the binding of both Delta and Omicron variants of SARS-CoV-2 to ACE2-expressing cells. In previous studies, the ability of AsA to decrease the cellular expression of ACE2 receptors in lung small airway epithelial cells (SAEC) and human microvascular endothelial cells (HMEC) at the protein and the RNA levels only occurred at high concentrations and after 6 days of cell exposure. AsA showed moderate additional benefits in decreasing ACE2 expression when combined with NAC [32]. Our data provide evidence that AsA does not further enhance the ability of NAC to inhibit the cellular uptake of SARS-CoV-2 pseudoviruses.

ROS plays an important role in the intracellular alterations produced by SARS-CoV-2, leading to the massive production of proinflammatory cytokines and chemokines, such as TNFα, IL6, and IL8, referred to as “cytokine storm”, which is responsible for lung tissue damage and causes cell death [33], and for ARDS, which involves a systemic inflammatory response that has been attributed to the release of mediators triggering an attack by the immune system [34]. Thus, also in COVID-19, there is an interplay between intracellular oxidative damage, inflammation, and immunopathological response [21,35]. In contrast with intracellular conditions, the extracellular milieu lacks efficient redox homeostasis systems [36], while this environment is characterized by the activity of oxidoreductases that sustain thiol-disulfide exchange processes [37]. Little is known about the influence of ROS on SARS-CoV-2 binding to ACE2. Our experiments, simulating possible effects of oxidants and antioxidants in the extracellular compartment, did not show any influence of ROS on the cellular uptake of SARS-CoV-2 pseudoviruses derived from either the Delta or Omicron variants. In addition, neither H_2_O_2_ nor an ROS-generating system interfered with the ability of NAC to inhibit that mechanism. In this context, it has been known for a long time that NAC, L-Cys, and GSH are potent scavengers of ROS and especially of hypochlorous acid (HOCl) and ^•^OH, and additionally of H_2_O_2_ [38], but apparently, under our experimental conditions, ROS did not affect the protective effect of NAC on the thiol–disulfide equilibrium in the extracellular environment. Therefore, our data do not support the hypothesis that increased levels of oxidation may influence the ACE-2 receptor-dependent mechanism of SARS-CoV-2 infection [25].

An advantage of NAC among thiol compounds is that this molecule has been in clinical use for 60 years, even at very high doses and by various administration routes. NAC works as a precursor of GSH inside cells, so that all its intracellular effects are mediated by GSH replenishment. GSH plays a central role as a defense system against COVID-19 and other infectious diseases, as well as against chronic degenerative diseases that may occur as comorbidities [31]. Our experimental conditions mimic the effects in the extracellular environment and accordingly they are presumably due to the NAC itself. They are comparable to a situation that may occur in cells of the nasal, mouth, and respiratory tract when the drug is administered either by intranasal instillation, by mouthwash, or by aerosol nebulization. It has been proposed that the nebulization of NAC may be a viable alternative for the management of the early stages of COVID-19 [39].

A number of authors have proposed the use of NAC in COVID-19 (see almost 100 citations in https://pubmed.ncbi.nlm.nih.gov/?term=acetylcysteine+and+covid-19&sort=date&size=20, accessed on 19 October 2022). In fact, NAC may play a role in the prevention of the disease as well as in its adjuvant therapy, both by counteracting the attack of SARS-CoV-2 and by mitigating COVID-19 infection. Clearly, such a preventive pharmacological approach would not be alternative but complementary to anti-COVID-19 vaccination. Both strategies target the SARS-CoV-2 spikes in such a way to hamper its binding to the ACE2 receptor. Vaccination has the advantage to provide a long-lasting immunological protection, but it has the disadvantage to require a time lag and multiple doses before acquiring immunity. Above all, the efficacy of vaccination can be compromised by the circumstance that SARS-CoV-2 is a highly mutable virus giving rise to new variants that may partially overcome the immunological shield. On the other hand, certain drugs, such as NAC, may act not only extracellularly but are potentially capable of inducing a variety of intracellular protective mechanisms [31]. Furthermore, they have the advantage to act soon after their administration, which may be useful in people having contacts with SARS-CoV-2 patients. At the same time, however, they have the disadvantage of possessing reversible mechanisms that require a continuous intake. Importantly, as shown in the present study evaluating both the Delta and Omicron variants, NAC may have public health benefits by decreasing the risk of developing COVID-19 irrespective of the virus variant. In addition, it may protect against other virus infections as well as against those chronic degenerative diseases, often associated with COVID-19, which may take benefit from GSH replenishment. To this respect, it is noteworthy that, in a clinical trial, the daily administration of oral NAC during the cold season was found to significantly attenuate the incidence of influenza and influenza-like illnesses [40].

## Figures and Tables

**Figure 1 cells-11-03313-f001:**
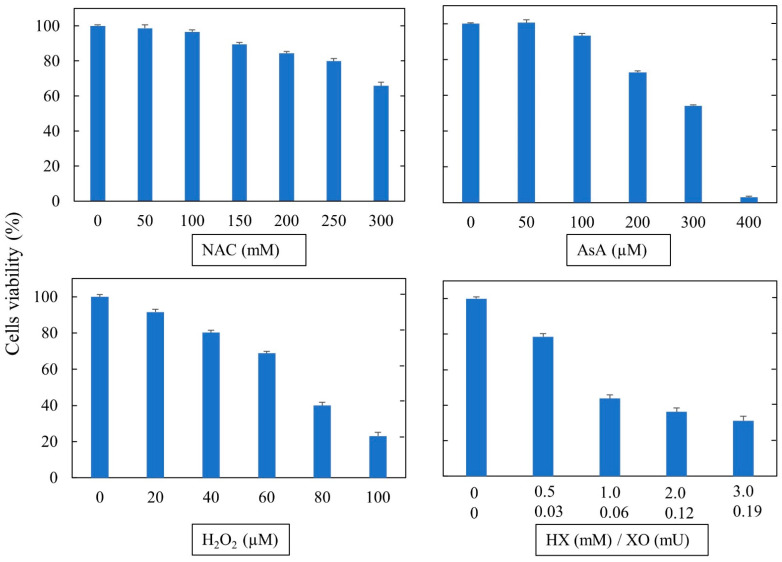
Evaluation of cytotoxicity of test compounds, as assessed by measuring PI incorporation in ACE2-HEK293 cells. The results are means ± SD of triplicates.

**Figure 2 cells-11-03313-f002:**
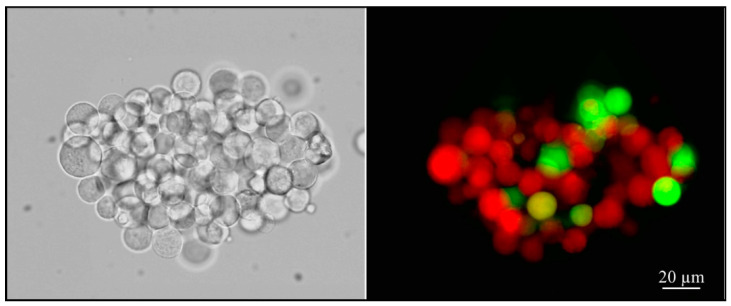
Representative photomicrograph of HEK293T cells transduction by internalization of pseudoviruses into ACE2-HEK293 cells. The enhanced green fluorescent protein (eGFP) is expressed upon integration into target cells (panel on the right). The fluorescence was recorded at 54 h post-transduction. Magnification 200×.

**Figure 3 cells-11-03313-f003:**
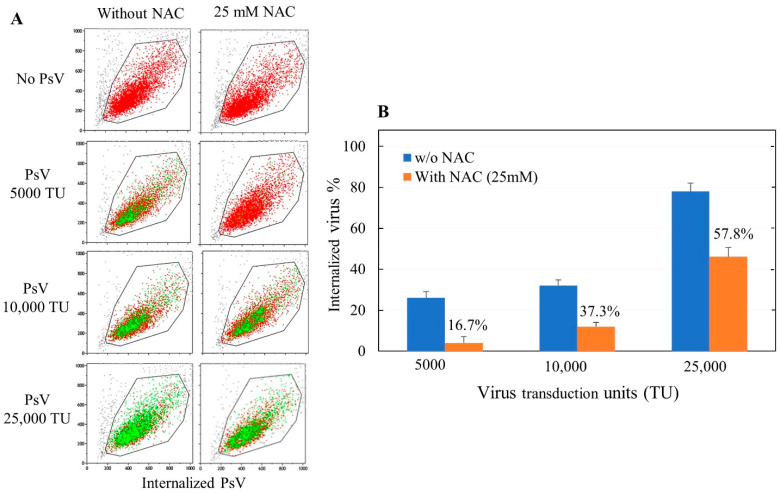
(**A**). Examples of cytofluorimetric analyses of ACE2-HEK293 cells, either unexposed or exposed to delta SARS-CoV-2 pseudovirus (PsV) at the doses of 5000, 10,000, and 25,000 TU/well, either in the absence or in the presence of 25 mM NAC. The red dots indicate pseudovirus-free cells, whereas the green dots indicate cells in which the pseudovirus was internalized. (**B**). Relationship between the virus dose and the proportion of virus internalized either in the absence or in the presence of NAC. The numbers above the NAC columns indicate, at each virus dose, the percent of internalized virus in the presence of NAC as related to the corresponding dose of internalized virus in the absence of NAC.

**Figure 4 cells-11-03313-f004:**
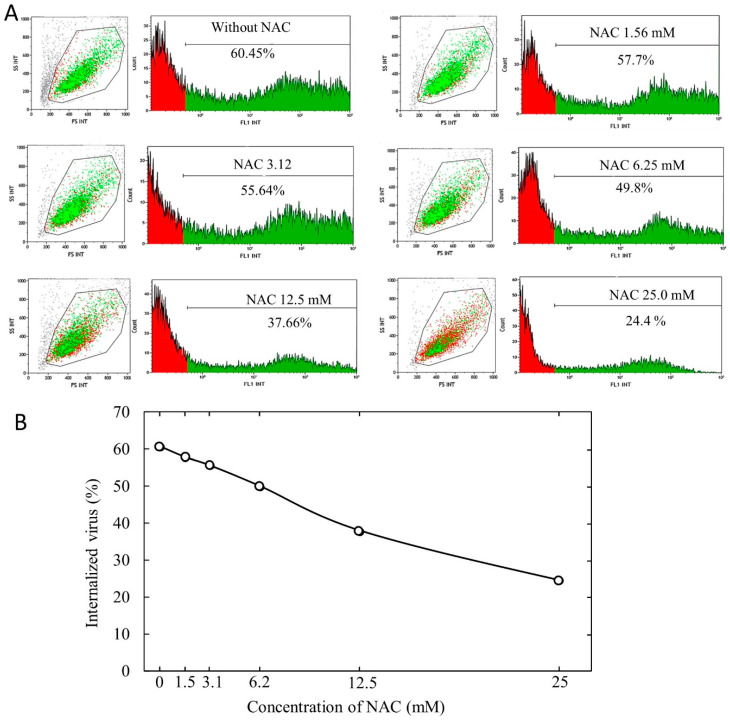
(**A**). Examples of cytofluorimetric analyses of ACE2-HEK293 cells exposed to delta SARS-CoV-2 pseudovirus (10,000 TU/culture), either in the absence or in the presence of varying concentrations of NAC, ranging between 1.56 and 12.5 mM. The green dots and areas in the cytofluorimetric plots reported at each NAC concentration indicate cells in which the pseudovirus was internalized, whereas the red dots and areas indicate pseudovirus-free cells. (**B**). Virus internalization curve as related to NAC concentration.

**Figure 5 cells-11-03313-f005:**
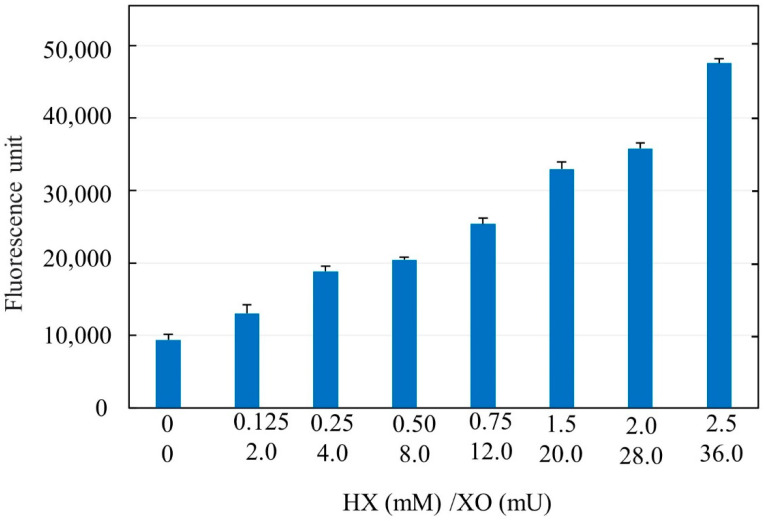
Dose-dependent generation of fluorescent ROS, as assessed by the DCFH method. The results are the means ± SD of triplicate analyses and are expressed as arbitrary fluorescence units.

**Figure 6 cells-11-03313-f006:**
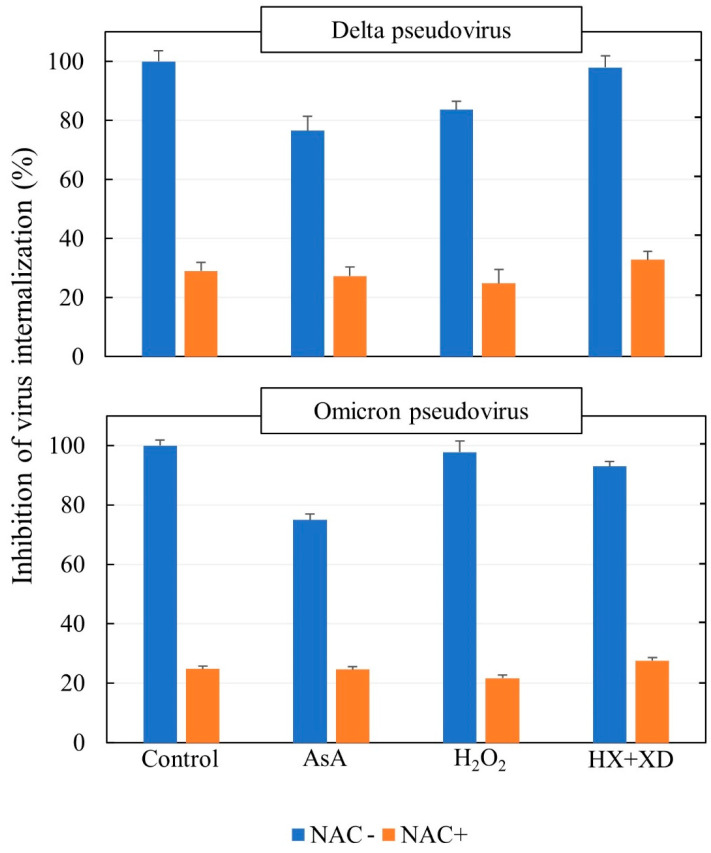
Internalization of Delta and Omicron SARS-CoV-2 pseudoviruses (10,000 TU/culture) into ACE2-HEK293 cells, as related to the treatment of cells with either ascorbic acid (AsA, 200 µM) or hydrogen peroxide (H_2_O_2_, 50 µM), or a mixture of hypoxanthine (HX, 31 µM) and xanthine oxidase (XO, 0.5 mU/mL), and either in the absence or in the presence of NAC (25 mM). The results are the means ± SD of triplicate analyses and are expressed as the percent of the values recorded in virus controls.

## Data Availability

The data presented in this study are available on request from the corresponding authors.

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
