# Peer review of "Inhibition of the Cell Uptake of Delta and Omicron SARS-CoV-2 Pseudoviruses by *N*-Acetylcysteine Irrespective of the Oxidoreductive Environment"

_cells, 2022, doi:10.3390/cells11203313_

Round 1

Reviewer 1 Report

The manuscript by Sebastiano La Maestra et. al., titled "Inhibition of the Cell Uptake of Delta and Omicron SARS-CoV-2 Pseudoviruses by N-Acetylcysteine Irrespective of the Oxidoreductive Environment" is a timely article touching on important aspect of COVID-19. 

The findings with respect to "explored the involvement of oxidoreductive mechanisms by investigating the effects of oxidants and antioxidants on virus uptake by ACE2 expressing cells of human origin (ACE2-HEK293). The cell uptake of pseudoviruses carrying the envelope of either delta or omicron variants of SARS-CoV-2 was evaluated by means of a cytofluorimetric approach. The thiol N-acetyl-L-cysteine (NAC) inhibited the uptake of both variants in a reproducible and dose-dependent fashion" are important findings.

At the same time, while evaluating it deeper, incorporation of few specific suggestions would augment it further. This includes,

1) The Figure 5 and 6 does not show standard deviation for the experimental replicates, highlighted by the barplots. That would be important to capture the variability within the replicates. 

2) Can the in-vitro findings be substantiated with patient samples, especially from the recent Omicron infected patients?

3) How would the authors factor in the role of vaccination, one of the critical difference between the Delta and Omicron led global wave of SARS-CoV-2 infections?

4) How it would augment public health benefits? Some aspects can be highlighted further.

Author Response

Author’s Reply to the Review Report (Reviewer 1)

1) As suggested, we added the SD values to the bars in Figures 5 and 6. We changed the figure legends accordingly.

2) The present one was a preclinical study, and the end-point investigated could mechanistically explored only under controlled experimental conditions. However, since the methodology used is now well set up in our laboratory, we are discussing the possibility of applying it in the future to investigate the ability of blood serum from either COVID-19 patients or vaccinated individuals to inhibit binding of delta and omicron pseudoviruses to ACE2 expressing cells.

3) and 4) We completely changed the last paragraph of the manuscript, also by adding a new reference, by comparing the roles of a pharmacological approach versus the advantages and disadvantages of vaccination. Moreover, we mentioned the public health benefits of the pharmacological approach. In particular, the last paragraph was revised and extended as follows:

“A number of authors have discussed the use of NAC in COVID-19 [see 100 citations in https://pubmed.ncbi.nlm.nih.gov/?term=acetylcysteine+and+covid-19&sort=date&size=20].  In fact, NAC may play a role in the prevention of the disease as well as in its adjuvant therapy both by counteracting the attack of SARS-CoV-2 and by mitigating COVID-19 infection.  Clearly, such a preventive pharmacological approach would not be alternative but complementary to anti-COVID-19 vaccination.  Both strategies target the SARS-CoV-2 spikes in such a way to hamper its binding to the ACE2 receptor.  Vaccination has the advantage to provide a long-lasting immunological protection but it has the disadvantage to require a time lag and multiple doses before acquiring immunity.  Above all, the efficacy of vaccination can be compromised by the circumstance that SARS-CoV-2 is a highly mutable virus giving rise to new variants that may partially overcome the immunological shield [40].  On the other hand, certain drugs, such as NAC, may act not only extracellularly but are potentially capable of inducing a variety of intracellular protective mechanisms [31].  Furthermore, they have the advantage to act soon after their administration, which may be useful in people having contacts with SARS-CoV-2 patients.  At the same time, however, they have the disadvantage of possessing reversible mechanisms that require a continuous intake.  Importantly, as shown in the present study evaluating both delta and omicron variants, NAC may have public health benefits by decreasing the risk of developing COVID-19 irrespective of the virus variant.  In addition, it may protect against other virus infections as well as against those chronic degenerative diseases, often associated with COVID-19, which may take benefit from GSH replenishment.  To this respect, it is noteworthy that the daily administration of oral NAC during the cold season was found to significantly attenuate the incidence of influenza and influenza-like illnesses [41].

Reviewer 2 Report

The study shown is of a high standard and interesting to readers. The analysis is thorough and that deserves publication in the current form.

The strength of this article is to show the mechanisms of action involving virus internalization by ACE2. Many papers in the literature discuss the importance of the ACE receptor but none has explained the exact mechanism so well. The weakness of this manuscript concerns the relative pharmacological application of what is stated. The possible role of NAC is mentioned but reference is made to other studies in the literature without elaborating further. However, this does not impact the success of the paper, which remains of absolute interest to readers.

Author Response

Author’s Reply to the Review Report (Reviewer 2)

We thank this Reviewer for his appreciation words and for stating that the study deserves publication in the current form.

Since he commented that “the weakness of this manuscript concerns the relative pharmacological application of what is stated”, the modified paragraph reported in the reply to Reviewer 1 addresses this issue.

Round 2

Reviewer 1 Report

Authors have satisfactorily addressed the suggestions to the best possible extent.